# Learning a Deep Compact Image Representation for Visual Tracking

**Naiyan Wang**     **Dit-Yan Yeung**
Department of Computer Science and Engineering
Hong Kong University of Science and Technology
winsty@gmail.com     dyyeung@cse.ust.hk

## Abstract

In this paper, we study the challenging problem of tracking the trajectory of a moving object in a video with possibly very complex background. In contrast to most existing trackers which only learn the appearance of the tracked object online, we take a different approach, inspired by recent advances in deep learning architectures, by putting more emphasis on the (unsupervised) feature learning problem. Specifically, by using auxiliary natural images, we train a stacked denoising autoencoder offline to learn generic image features that are more robust against variations. This is then followed by knowledge transfer from offline training to the online tracking process. Online tracking involves a classification neural network which is constructed from the encoder part of the trained autoencoder as a feature extractor and an additional classification layer. Both the feature extractor and the classifier can be further tuned to adapt to appearance changes of the moving object. Comparison with the state-of-the-art trackers on some challenging benchmark video sequences shows that our deep learning tracker is more accurate while maintaining low computational cost with real-time performance when our MATLAB implementation of the tracker is used with a modest graphics processing unit (GPU).

## 1   Introduction

Visual tracking, also called object tracking, refers to automatic estimation of the trajectory of an object as it moves around in a video. It has numerous applications in many domains, including video surveillance for security, human-computer interaction, and sports video analysis. While a certain application may require multiple moving objects be tracked, the typical setting is to treat each object separately. After the object to track is identified either manually or automatically in the first video frame, the goal of visual tracking is to automatically track the trajectory of the object over the subsequent frames. Although existing computer vision techniques may offer satisfactory solutions to this problem under well-controlled environments, the problem can be very challenging in many practical applications due to factors such as partial occlusion, cluttered background, fast and abrupt motion, dramatic illumination changes, and large variations in viewpoint and pose.

Most existing trackers adopt either the generative or the discriminative approach. Generative trackers, like other generative models in machine learning, assume that the object being tracked can be described by some generative process and hence tracking corresponds to finding the most probable candidate among possibly infinitely many. The motivation behind generative trackers is to develop image representations which can facilitate robust tracking. They have been inspired by recent advances in fast algorithms for robust estimation and sparse coding, such as the alternating direction method of multipliers (ADMM) and accelerated gradient methods. Some popular generative trackers include *incremental visual tracking* (IVT) [18], which represents the tracked object based on principal component analysis (PCA), and the $l_1$ *tracker* (L1T) [16], which assumes

that the tracked object can be represented by a sparse combination of overcomplete basis vectors. Many extensions [26, 25, 4, 21] have also been proposed. On the other hand, the discriminative approach treats tracking as a binary classification problem which learns to explicitly distinguish the object being tracked from its background. Some representative trackers in this category are the *online AdaBoost* (OAB) tracker [6], *multiple instance learning* (MIL) tracker [3], and *structured output tracker* (Struck) [8]. While generative trackers usually produce more accurate results under less complex environments due to the richer image representations used, discriminative trackers are more robust against strong occlusion and variations since they explicitly take the background into consideration. We refer the reader to a recent paper [23] which empirically compares many existing trackers based on a common benchmark.

From the learning perspective, visual tracking is challenging because it has only one labeled instance in the form of an identified object in the first video frame. In the subsequent frames, the tracker has to learn variations of the tracked object with only unlabeled data available. With no prior knowledge about the object being tracked, it is easy for the tracker to drift away from the target. To address this problem, some trackers taking the semi-supervised learning approach have been proposed [12, 7]. An alternative approach [22] first learns a dictionary of image features (such as SIFT local descriptors) from auxiliary data and then transfers the knowledge learned to online tracking.

Another issue is that many existing trackers make use of image representations that may not be good enough for robust tracking in complex environments. This is especially the case for discriminative trackers which usually put more emphasis on improving the classifiers rather than the image features used. While many trackers simply use raw pixels as features, some attempts have used more informative features, such as Haar features, histogram features, and local binary patterns. However, these features are all handcrafted offline but not tailor-made for the tracked object. Recently, deep learning architectures have been used successfully to give very promising results for some complicated tasks, including image classification [14] and speech recognition [10]. The key to success is to make use of deep architectures to learn richer invariant features via multiple nonlinear transformations. We believe that visual tracking can also benefit from deep learning for the same reasons.

In this paper, we propose a novel *deep learning tracker* (DLT) for robust visual tracking. We attempt to combine the philosophies behind both generative and discriminative trackers by developing a robust discriminative tracker which uses an effective image representation learned automatically. There are some key features which distinguish DLT from other existing trackers. First, it uses a *stacked denoising autoencoder* (SDAE) [20] to learn generic image features from a large image dataset as auxiliary data and then transfers the features learned to the online tracking task. Second, unlike some previous methods which also learn features from auxiliary data, the learned features in DLT can be further tuned to adapt to specific objects during the online tracking process. Because DLT makes use of multiple nonlinear transformations, the image representations obtained are more expressive than those of previous methods based on PCA. Moreover, since representing the tracked object does not require solving an optimization problem as in previous trackers based on sparse coding, DLT is significantly more efficient and hence is more suitable for real-time applications.

## 2 Particle Filter Approach for Visual Tracking

The particle filter approach [5] is commonly used for visual tracking. From the statistical perspective, it is a sequential Monte Carlo importance sampling method for estimating the latent state variables of a dynamical system based on a sequence of observations. Supppse $\mathbf{s}^t$ and $\mathbf{y}^t$ denote the latent state and observation variables, respectively, at time $t$. Mathematically, object tracking corresponds to the problem of finding the most probable state for each time step $t$ based on the observations up to the previous time step:

$$\begin{aligned}
\mathbf{s}^t &= \arg\max p(\mathbf{s}^t \mid \mathbf{y}^{1:t-1}) \\
&= \arg\max \int p(\mathbf{s}^t \mid \mathbf{s}^{t-1}) \, p(\mathbf{s}^{t-1} \mid \mathbf{y}^{1:t-1}) \, d\mathbf{s}^{t-1}.
\end{aligned} \tag{1}$$

When a new observation $\mathbf{y}^t$ arrives, the posterior distribution of the state variable is updated according to Bayes' rule:

$$p(\mathbf{s}^t \mid \mathbf{y}^{1:t}) = \frac{p(\mathbf{y}^t \mid \mathbf{s}^t) \, p(\mathbf{s}^t \mid \mathbf{y}^{1:t-1})}{p(\mathbf{y}^t \mid \mathbf{y}^{1:t-1})}. \tag{2}$$

What is specific to the particle filter approach is that it approximates the true posterior state distribution $p(\mathbf{s}^t \mid \mathbf{y}^{1:t})$ by a set of $n$ samples, called particles, $\{\mathbf{s}_i^t\}_{i=1}^n$ with corresponding importance weights $\{w_i^t\}_{i=1}^n$ which sum to 1. The particles are drawn from an importance distribution $q(\mathbf{s}^t \mid \mathbf{s}^{1:t-1}, \mathbf{y}^{1:t})$ and the weights are updated as follows:

$$w_i^t = w_i^{t-1} \cdot \frac{p(\mathbf{y}^t \mid \mathbf{s}_i^t)\, p(\mathbf{s}_i^t \mid \mathbf{s}_i^{t-1})}{q(\mathbf{s}^t \mid \mathbf{s}^{1:t-1}, \mathbf{y}^{1:t})}. \tag{3}$$

For the choice of the importance distribution $q(\mathbf{s}^t \mid \mathbf{s}^{1:t-1}, \mathbf{y}^{1:t})$, it is often simplified to a first-order Markov process $q(\mathbf{s}^t \mid \mathbf{s}^{t-1})$ in which state transition is independent of the observation. Consequently, the weights are updated as $w_i^t = w_i^{t-1} p(\mathbf{y}^t \mid \mathbf{s}_i^t)$. Note that the sum of weights may no longer be equal to 1 after each weight update step. In case it is smaller than a threshold, resampling is applied to draw $n$ particles from the current particle set in proportion to their weights and then resetting their weights to $1/n$. If the weight sum is above the threshold, linear normalization is applied to ensure that the weights sum to 1.

For object tracking, the state variable $\mathbf{s}_i$ usually represents the six affine transformation parameters which correspond to translation, scale, aspect ratio, rotation, and skewness. In particular, each dimension of $q(\mathbf{s}^t \mid \mathbf{s}^{t-1})$ is modeled independently by a normal distribution. For each frame, the tracking result is simply the particle with the largest weight. While many trackers also adopt the same particle filter approach, the main difference lies in the formulation of the observation model $p(\mathbf{y}^t \mid \mathbf{s}_i^t)$. Apparently, a good model should be able to distinguish well the tracked object from the background while still being robust against various types of object variation. For discriminative trackers, the formulation is often to set the probability exponentially related to the confidence of the classifier output.

The particle filter framework is the dominant approach in visual tracking for several reasons. First, it is more general than the Kalman filter approach by going beyond the Gaussian distribution. Moreover, it approximates the posterior state distribution by a set of particles instead of just a single point such as the mode. For visual tracking, this property makes it easier for the tracker to recover from incorrect tracking results. A tutorial on using particle filters for visual tracking can be found in [2]. Some recent work, e.g., [15], further improves the particle filter framework for visual tracking.

## 3 The DLT Tracker

We now present our DLT tracker. During the offline training stage, unsupervised feature learning is carried out by training an SDAE with auxiliary image data to learn generic natural image features. Layer-by-layer pretraining is first applied and then the whole SDAE is fine-tuned. During the online tracking process, an additional classification layer is added to the encoder part of the trained SDAE to result in a classification neural network. More details are provided in the rest of this section.

### 3.1 Offline Training with Auxiliary Data

#### 3.1.1 Dataset and Preprocessing

We use the Tiny Images dataset [19] as auxiliary data for offline training. The dataset was collected from the web by providing non-abstract English nouns to seven search engines, covering many of the objects and scenes found in the real world. From the almost 80 million tiny images each of size $32 \times 32$, we randomly sample 1 million images for offline training. Since most state-of-the-art trackers included in our empirical comparison use only grayscale images, we have converted all the sampled images to grayscale (but our method can also use the color images directly if necessary). Consequently, each image is represented by a vector of 1024 dimensions corresponding to 1024 pixels. The feature value of each dimension is linearly scaled to the range $[0, 1]$ but no further preprocessing is applied.

#### 3.1.2 Learning Generic Image Features with a Stacked Denoising Autoencoder

The basic building block of an SDAE is a one-layer neural network called a *denoising autoencoder* (DAE), which is a more recent variant of the conventional autoencoder. It learns to recover a data sample from its corrupted version. In so doing, robust features are learned since the neural network

contains a "bottleneck" which is a hidden layer with fewer units than the input units. We show the architecture of DAE in Fig. 1(a).

Let there be a total of $k$ training samples. For the $i$th sample, let $\mathbf{x}_i$ denote the original data sample and $\tilde{\mathbf{x}}_i$ be the corrupted version of $\mathbf{x}_i$, where the corruption could be masking corruption, additive Gaussian noise or salt-and-pepper noise. For the network weights, let $\mathbf{W}$ and $\mathbf{W}'$ denote the weights for the encoder and decoder, respectively, which may be tied though it is not necessary. Similarly, $\mathbf{b}$ and $\mathbf{b}'$ refer to the bias terms. A DAE learns by solving the following (regularized) optimization problem:

$$\min_{\mathbf{W},\mathbf{W}',\mathbf{b},\mathbf{b}'} \sum_{i=1}^{k} \|\mathbf{x}_i - \hat{\mathbf{x}}_i\|_2^2 + \lambda(\|\mathbf{W}\|_F^2 + \|\mathbf{W}'\|_F^2), \tag{4}$$

where

$$\begin{aligned} \mathbf{h}_i &= f(\mathbf{W}\tilde{\mathbf{x}}_i + \mathbf{b}) \\ \hat{\mathbf{x}}_i &= f(\mathbf{W}'\mathbf{h}_i + \mathbf{b}'). \end{aligned} \tag{5}$$

Here $\lambda$ is a parameter which balances the reconstruction loss and weight penalty terms, $\|\cdot\|_F$ denotes the Frobenius norm, and $f(\cdot)$ is a nonlinear activation function which is typically the logistic sigmoid function or hyperbolic tangent function. By reconstructing the input from a corrupted version of it, a DAE is more effective than the conventional autoencoder in discovering more robust features by preventing the autoencoder from simply learning the identity mapping.

To further enhance learning meaningful features, sparsity constraints [9] are imposed on the mean activation values of the hidden units. If the logistic sigmoid activation function is used, the output of each unit may be regarded as the probability of it being active. Let $\rho_j$ denote the target sparsity level of the $j$th unit and $\hat{\rho}_j$ its average empirical activation rate. The cross-entropy of $\boldsymbol{\rho}$ and $\hat{\boldsymbol{\rho}}$ can then be introduced as an additional penalty term to Eqn. 4:

$$\begin{aligned} H(\boldsymbol{\rho} \,\|\, \hat{\boldsymbol{\rho}}) &= -\sum_{j=1}^{m} \Big[ \rho_j \log(\hat{\rho}_j) + (1 - \rho_j) \log(1 - \hat{\rho}_j) \Big] \\ \hat{\boldsymbol{\rho}} &= \frac{1}{k} \sum_{i=1}^{k} \mathbf{h}_i, \end{aligned} \tag{6}$$

where $m$ is the number of hidden units. After the pretraining phase, the SDAE can be unrolled to form a feedforward neural network. The whole network is fine-tuned using the classical backpropagation algorithm. To increase the convergence rate, either the simple momentum method or more advanced optimization techniques such as the L-BFGS or conjugate gradient method can be applied.

For the network architecture, we use overcomplete filters in the first layer. This is a deliberate choice since it has been found that an overcomplete basis can usually capture the image structure better. This is in line with the neurophysiological mechanism in the V1 visual cortex [17]. Then the number of units is reduced by half whenever a new layer is added until there are only 256 hidden units, serving as the bottleneck of the autoencoder. The whole structure of the SDAE is depicted in Fig. 1(b). To further speed up pretraining in the first layer to learn localized features, we divide each $32 \times 32$ tiny image into five $16 \times 16$ patches (upper left, upper right, lower left, lower right, and the center one which overlaps with the other four), and then train five DAEs each of which has 512 hidden units. After that, we initialize a large DAE with the weights of the five small DAEs and then train the large DAE normally. Some randomly selected filters in the first layer are shown in Fig. 2. As expected, most of the filters play the role of highly localized edge detectors.

### 3.2 Online Tracking Process

The object to track is specified by the location of its bounding box in the first frame. Some negative examples are collected from the background at a short distance from the object. A sigmoid classification layer is then added to the encoder part of the SDAE obtained from offline training. The overall network architecture is shown in Fig. 1(c). When a new video frame arrives, we first draw particles according to the particle filter approach. The confidence $p_i$ of each particle is then determined by making a simple forward pass through the network. An appealing characteristic of this approach is that the computational cost of this step is very low even though it has high accuracy.

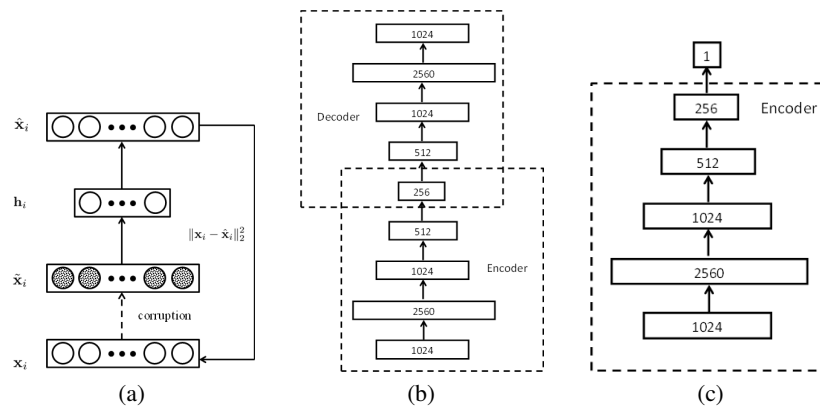

Figure 1: Some key components of the network architecture: (a) denoising autoencoder; (b) stacked denoising autoencoder; (c) network for online tracking.

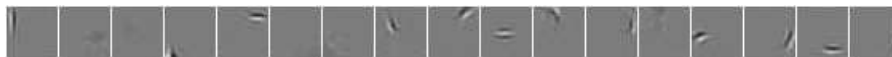

Figure 2: Some filters in the first layer of the learned SDAE.

If the maximum confidence of all particles in a frame is below a predefined threshold $\tau$, it may indicate significant appearance change of the object being tracked. To address this issue, the whole network can be tuned again in case this happens. We note that the threshold $\tau$ should be set by maintaining a tradeoff. If $\tau$ is too small, the tracker cannot adapt well to appearance changes. On the other hand, if $\tau$ is too large, even an occluding object or the background may be mis-treated as the tracked object and hence leads to drifting of the target.

## 4   Experiments

We empirically compare DLT with some state-of-the-art trackers in this section using 10 challenging benchmark video sequences. These trackers are: MTT [26], CT [24], VTD [15], MIL [3], a latest variant of L1T [4], TLD [13], and IVT [18]. We use the original implementations of these trackers provided by their authors. In case a tracker can only deal with grayscale video, the `rgb2gray` function provided by the MATLAB Image Processing Toolbox is used to convert the color video to grayscale. To accelerate the computation, we also utilize GPU computation provided by the MATLAB Parallel Computing Toolbox in both offline training and online tracking. The codes and supplemental material are provided on the project page: `http://winsty.net/dlt.html`.

### 4.1   DLT Implementation Details

We use the gradient method with momentum for optimization. The momentum parameter is set to 0.9. For offline training of the SDAE, we inject Gaussian noise with a variance of 0.0004 to generate the corrupted input. We set $\lambda = 0.0001$, $\rho_i = 0.05$, and the mini-batch size to 100. For online tuning, we use a larger $\lambda$ value of 0.002 to avoid overfitting and a smaller mini-batch size of 10. The threshold $\tau$ is set to 0.9. The particle filter uses 1000 particles. For other parameters such as the affine parameters in the particle filter and the search window size in the other methods, we perform grid search to determine the best values. The same setting is applied to all other methods compared if applicable.

### 4.2   Quantitative Comparison

We use two common performance metrics for quantitative comparison: success rate and central-pixel error. Let $BB_T$ denote the bounding box produced by a tracker and $BB_G$ the ground-truth

bounding box. For each video frame, a tracker is considered successful if the overlap percentage $\frac{\text{area}(BB_T \cap BB_G)}{\text{area}(BB_T \cup BB_G)} > 50\%$. As for the central-pixel error, it is defined as the Euclidean distance (in pixels) between the centers of $BB_T$ and $BB_G$. The quantitative comparison results are summarized in Table 1 . For each row which corresponds to one of 10 video sequences, the best result is shown in red and second best in blue. We also report the central-pixel errors over all frames for each video sequence. Since TLD can report that the tracked object is missing in some frames, we exclude it from the central-pixel error comparison. On average, DLT is the best according to both performance metrics. For most video sequences, it is among the best two methods. We also list the running time of each sequence in detail in Table 2. Thanks to advances of the GPU technology, our tracker can achieve an average frame rate of 15fps (frames per second) which is sufficient for many real-time applications.

|  | Ours | MTT | CT | VTD | MIL | L1T | TLD | IVT |
|---|---|---|---|---|---|---|---|---|
| car4 | 100(6.0) | 100(3.4) | 24.7(95.4) | 35.2(41.5) | 24.7(81.8) | 30.8(16.8) | 0.2(-) | 100(4.2) |
| car11 | 100(1.2) | 100(1.3) | 70.7(6.0) | 65.6(23.9) | 68.4(19.3) | 100(1.3) | 29.8(-) | 100(3.2) |
| davidin | 66.1(7.1) | 68.6(7.8) | 25.3(15.3) | 49.4(27.1) | 17.7(13.1) | 27.3(17.5) | 44.4(-) | 92.0(3.9) |
| trellis | 93.6(3.3) | 66.3(33.7) | 23.0(80.4) | 30.1(81.3) | 25.9(71.7) | 62.1(37.6) | 48.9(-) | 44.3(44.7) |
| woman | 67.1(9.4) | 19.8(257.8) | 16.0(109.6) | 17.1(133.6) | 12.2(123.7) | 21.1(138.2) | 5.8(-) | 21.5(111.2) |
| animal | 87.3(10.2) | 88.7(11.1) | 85.9(10.8) | 91.5(10.8) | 63.4(16.1) | 85.9(10.4) | 63.4(-) | 81.7(10.8) |
| shaking | 88.4(11.5) | 12.3(28.1) | 92.3(10.9) | 99.2(5.2) | 26.0(28.6) | 0.5(90.8) | 15.6(-) | 1.1(138.4) |
| singer1 | 100(3.3) | 35.6(34.0) | 10.3(16.8) | 99.4(3.4) | 10.3(26.0) | 100(3.7) | 53.6(-) | 96.3(7.9) |
| surfer | 86.5(4.6) | 83.8(6.9) | 13.5(18.7) | 90.5(5.5) | 44.6(14.7) | 75.7(9.5) | 40.5(-) | 90.5(5.9) |
| bird2 | 65.9(16.8) | 9.2(92.8) | 58.2(19.7) | 13.3(151.1) | 69.4(16.3) | 45.9(57.5) | 31.6(-) | 10.2(104.1) |
| average | 85.5(7.3) | 58.4(47.6) | 42.0(38.4) | 59.1(48.4) | 36.3(41.1) | 54.9(40.1) | 33.4(-) | 63.8(43.4) |

Table 1: Comparison of 8 trackers on 10 video sequences. The first number denotes the success rate (in percentage), while the number in parentheses denotes the central-pixel error (in pixels).

| car4 | car11 | davidin | trellis | woman | animal | shaking | singer1 | surfer | bird2 | Average |
|---|---|---|---|---|---|---|---|---|---|---|
| 15.27 | 16.04 | 13.20 | 17.30 | 20.92 | 10.93 | 12.72 | 15.18 | 14.17 | 14.36 | 15.01 |

Table 2: Comparison of running time on 10 video sequences (in fps).

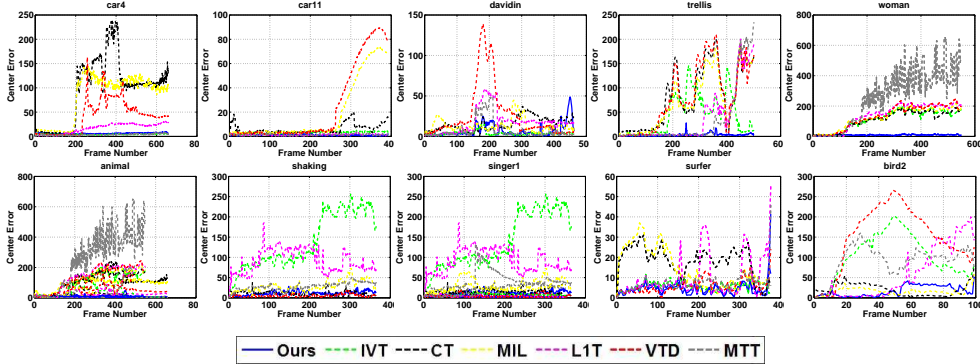

Figure 3: Frame-by-frame comparison of 7 trackers on 10 video sequences in terms of central-pixel error (in pixels).

### 4.3 Qualitative Comparison

Fig. 4 shows some key frames with bounding boxes reported by all eight trackers for each of the 10 video sequences. More detailed results for the complete video sequences can be found in the supplemental material.

In both the *car4* and *car11* sequences, the tracked objects are cars moving on an open road. For *car4*, the challenge is that the illumination changes greatly near the entrance of a tunnel. For *car11*, the

environment is very dark with illumination in the cluttered background. Since the car being tracked is a rigid object, its shape does not change much and hence generative trackers like IVT, L1T and MTT generally perform well for these two sequences. DLT can also track the car accurately.

In the *davidin* and *trellis* sequences, each tracker has to track a face in indoor and outdoor environments, respectively. Both sequences are challenging because the illumination and pose vary drastically along the video. Moreover, out-of-plane rotation occurs in some frames. As a consequence, all trackers drift or even fail to different degrees. Generally speaking, DLT and MTT yield the best results which are followed by IVT.

In the *woman* sequence, we track a woman walking in the street. The woman is severely occluded several times by the parked cars. TLD first fails at frame 63 because of the pose change. All other trackers compared fail when the woman walks close to the car at about frame 130. DLT can follow the target accurately.

In the *animal* sequence, the target is a fast moving animal with motion blur. All methods can merely track the target to the end. Only MIL and TLD fail in some frames. TLD is also misled by some similar objects in the background, e.g., in frame 41.

Both the *shaking* and *singer1* sequences are recordings on the stage with illumination changes. For *shaking*, the pose of the head being tracked also changes. L1T, IVT and TLD totally fail before frame 10, while MTT and MIL show some drifting effects then. VTD and DLT give satisfactory results which are followed by CT. Compared to *shaking*, the *singer1* sequence is easier to track. All trackers except MTT can track the object but CT and MIL do not support scale change and hence the results are less satisfactory.

In the *surfer* sequence, the goal is to track the head of a surfer while its pose changes along the video sequence. All trackers can merely track it except that TLD shows an incorrect scale and both CT and MIL drift slightly.

The *bird2* sequence is very challenging since the pose of the bird changes drastically when it is occluded. Most trackers fail or drift at about frame 15 with the exception of L1T, TLD and DLT. However, after the bird turns, L1T and TLD totally fail but CT and MIL can recover to some degree. DLT can track the bird accurately along the entire sequence.

## 5 Discussions

Our proposed method is similar in spirit to that of [22] but there are some key differences that are worth noting. First, we learn generic image features from a larger and more general dataset rather than a smaller set with only some chosen image categories. Second, we learn the image features from raw images automatically instead of relying on handcrafted SIFT features. Third, further learning is allowed during the online tracking process of our method so as to adapt better to the specific object being tracked.

For the choice of deep network architecture, we note that another potential candidate is the popular *convolutional neural network* (CNN) model. The resulting tracker would be similar to previous patch (or fragment) based methods [1, 11] which have been shown to be robust against partial occlusion. Nevertheless, current research on CNN focuses on learning *shift-invariant* features for such tasks as image classification and object detection. However, the nature of object tracking is very different in that it has to learn *shift-variant* but similarity-preserving features to overcome the drifting problem. As of now, there is very little relevant work, with the possible exception of [11] which tries to improve the pooling step in the sparse coding literature to address this issue. This could be an interesting future research direction to pursue.

## 6 Concluding Remarks

In this paper, we have successfully taken deep learning to a new territory of challenging applications. Noting that the key to success for deep learning architectures is the learning of useful features, we first train a stacked denoising autoencoder using many auxiliary natural images to learn generic image features. This alleviates the problem of not having much labeled data in visual tracking applications. After offline training, the encoder part of the SDAE is used as a feature extractor

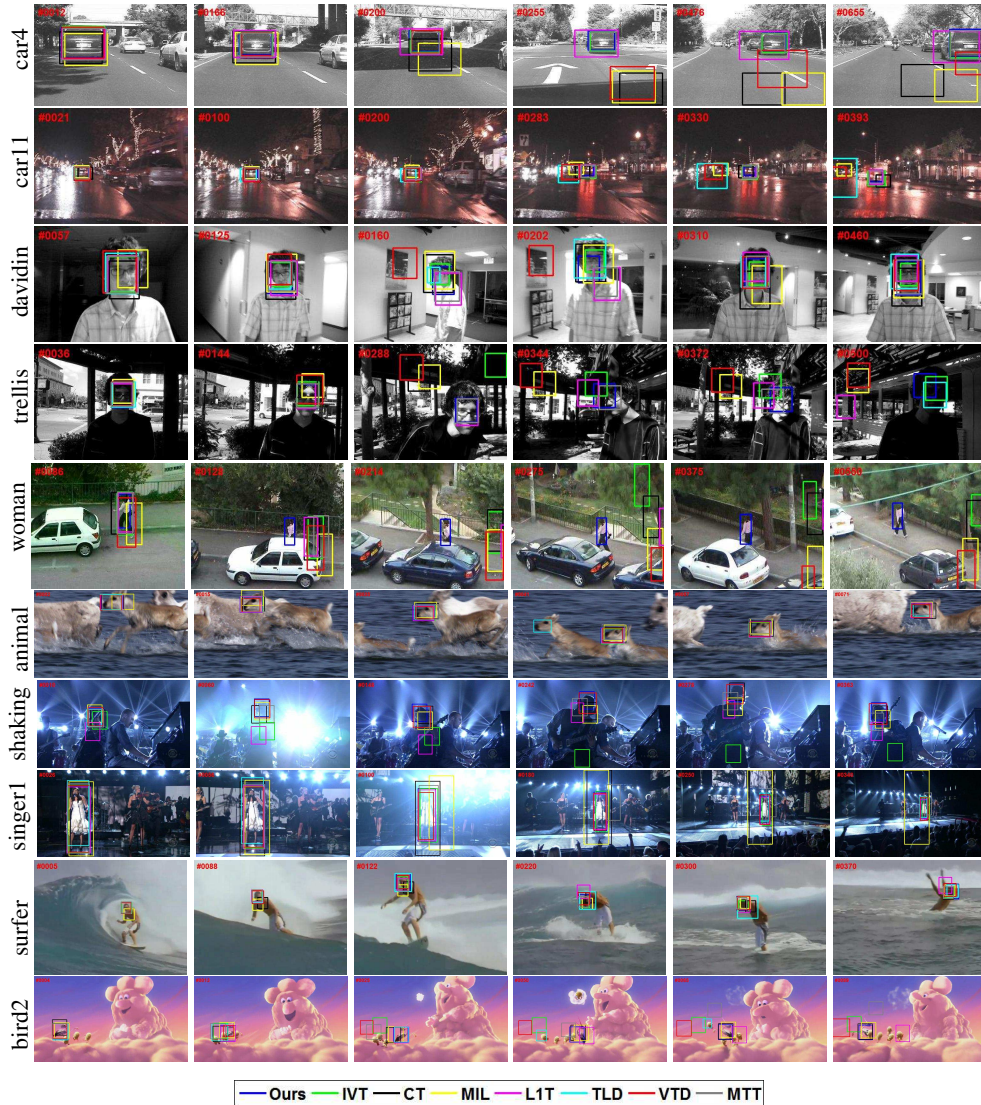

Figure 4: Comparison of 8 trackers on 10 video sequences in terms of the bounding box reported.

during the online tracking process to train a classification neural network to distinguish the tracked object from the background. This can be regarded as knowledge transfer from offline training using auxiliary data to online tracking. Since further tuning is allowed during the online tracking process, both the feature extractor and the classifier can adapt to appearance changes of the moving object. Through quantitative and qualitative comparison with state-of-the-art trackers on some challenging benchmark video sequences, we demonstrate that our deep learning tracker gives very encouraging results while having low computational cost.

As the first work on applying deep neural networks to visual tracking, many opportunities remain open for further research. As discussed above, it would be an interesting direction to investigate a shift-variant CNN. Also, the classification layer in our current tracker is just a linear classifier for simplicity. Extending it to more powerful classifiers, as in other discriminative trackers, may provide more room for further performance improvement.

## Acknowledgment

This research has been supported by General Research Fund 621310 from the Research Grants Council of Hong Kong.

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
