[Reviews · NeurIPS 2013]

Submitted by Assigned_Reviewer_1

The paper present the use of off-line deep learning approach as a method for improving online visual tracking.
The method trains a neural network for the representation "generic" images.

The paper is well written and can be followed easily. The Offline approach is well explained, however there are lack of explanations for the online tracking. It is not clear enough how it is exactly done.

The paper is evaluated against the the baselines in online tracking, and present promising results.
The authors themselves claim that their paper is similar in spirit to [22] it would be interesting to compare the tracking results.
In the paper [22] there are many publicly available datasets that can be used for the comparison.
Interestingly, the authors use the same evaluation metric as [22] but non of the scores is similar to the table of [22] (for example for the "shaking" sequence) - could the authors comment on that?

In vision, DLT is usually stands for: Direct linear transformation (calibration, etc'.)
As this is a paper for visual tracking, I strongly recommend to change the abbreviation of the algorithm.

In the online tracking domain, the code of the state-of-the-art methods is publicly available. Do the authors intend to release the code as well?
Summary: The paper present the use of off-line deep learning approach as a method for improving online visual tracking.
The paper is well written and can be followed easily. The paper is evaluated against the the baselines in online tracking, and present promising results.

I recommend to accept the paper.

Submitted by Assigned_Reviewer_8

This paper presents an application of deep learning to visual tracking. Under the framework of particle filter, this paper builds a likelihood model based on learned image representation from stacked denoising auto-encoder (SDAE), and achieves good results on 11 selected videos compared to previous trackers. The reviewer will evaluate this paper from four aspects below.

1. Quality
The main algorithms (particle filter and SDAE) used in this paper are well known and should be correct. Unsupervised feature learning is also well established for image representation. There is no surprise when it works reasonably well for representing tracking targets. It will be good that the authors can present some insights that how the deep learning here help solve any intrinsic problems of visual tracking? For example, is the learned representation robust to occlusions and motion blur? Do the learned features make the visual matching easier, or tracking targets more salient? In addition, the SDAE is online updated while tracking. The reviewer would like to know if this will cause over-fitting and thus tracking drift.

2. Clarity
This paper is clearly written and easy to read. Since the source codes of SDAE and particle filter are public available, the results should be easily reproduced.

3. Originality
This paper is an incremental work. It shares similar ideas and algorithm framework with [22] of combining feature learning with particle filter. The only difference is that this paper use SDAE for feature learning while [22] uses sparse coding for feature extraction. Although this paper learns features from pixels while [22] learns features from SIFT, no experimental results show the advantages of SDAE in this paper over [22]. The reviewer suggests the authors present thorough comparisons with [22] so as to make explicit the merits of the proposed algorithm.

4. Significance
Since the SDAE and particle filter are well known in the community, the direct combination of them bring little new insights towards both deep learning and visual tracking.
Summary: In summary, this paper looks like a solid work but its scientific quality is limited.

Submitted by Assigned_Reviewer_9

Summary.

The paper describes and studies the use of a stacked denoising autoencoder to solve the problem of object tracking in videos. The autoencoder is trained on auxiliary data (natural images), to produce “generic” features. These features are then used as the input of a neural network which is trained specifically for the task of tracking.

Quality.

The paper is nicely written and structured. The problem well exposed and easy to understand. The method clearly exposed, and results are provided for the main standard tracking benchmarks.

Clarity.

Very clear:
(1) the stacked auto encoder is trained on offline data (1M images from the tiny images dataset)
(2) a classifier is then trained online, using a particle filter to sample examples (the input to the classifier is the output of the auto encoder)
(3) if none of the predictions on a new frame are satisfying, the encoder can be fully refined / tuned, to produce better scores (this offers greater modeling capacity than a simple linear model)

Originality.

The idea of learning generic features (using deep learning or not) as the basis of an online learning system, especially for computer vision tasks, is not particularly new. But I guess the novelty here is the use of a stacked denoising autoencoder specifically.
Summary: A good paper that uses a mix of offline unsupervised learning and online tuning / refinement for the task of visual tracking.

Submitted by Assigned_Reviewer_12

This paper proposes an algorithm for visual tracking of single objects using deep learning of the object's appearance. The algorithm proceeds in two stages: First, using 1 mio images from the Tiny image dataset generic features for natural images are learnt using a stacked denoising autoencoder (SDA). Next, an extra classification layer is added that is learnt from the initial frame and updated over time as soon as detection fails. The observations from this appearance model are integrated into a particle filtering framework.

The paper is very well written and clear. Almost all parameters are well described. The paper seems to be novel to me with respect to combining deep learning architectures with tracking (though I am not 100% sure here). The results look good to me.

Points, that could help to improve the paper:

- The authors claim that their method is efficient, though a detailed listing of running times wrt. to evaluation and re-training (which has to occur quite often I guess) is missing.

- It is unclear if only parts or the whole network is re-trained online. Is it really the case that the authors are able to retrain the whole network at 2-2fps?

- It will be great if the authors could elaborate a bit more on the drift issue and the related parameter. What is the actual value of tau from the specified set?
Summary: This paper nicely combines deep learning architectures and visual object tracking, which, to the best of my knowledge has not been tried before.
Author Feedback

Author rebuttal: To reviewer 1:
Q: Their paper is similar in spirit to [22] it would be interesting to compare the tracking results.
A: Thanks for the suggestion. Since the authors of [22] do not make their code available, direct comparison with their results is not possible. We will try to implement their method by ourselves.

Q: The results are quite different from the results in [22].
A: Without sufficient details of the experiments reported in [22], it is hard to reproduce their results. Some possible explanations for the differences include: 1) difference in initial bounding box; 2) uncertainty caused by particle filtering framework; 3) different parameter settings.

Q: Do the authors intend to release the code as well?
A: Yes, as said in the first paragraph of Section 4, we will definitely make our implementation as well as details of the experiments available to help others reproduce our results.

To reviewer 2:
Q: Running time is missing.
A: The running time varies from case to case and it depends on how frequent fine tuning is applied. We will include some timing statistics in the revised version of our paper.

Q: It is unclear if only parts or the whole network is re-trained online. Is it really the case that the authors are able to retrain the whole network at 2-20fps?
A: Both the encoder and the classification layer are fine-tuned online (Fig 1(c)). Since the network has already been pre-trained, it typically needs no more than 5 iterations of backpropagation to fine-tune the network online in order to adapt to appearance changes. This explains why the reported speed could be achieved.

Q: If the authors could elaborate a bit more on the drift issue and what is the actual value of tau from the specified set?
A: The tunable parameter tau is set according to the speed of appearance changes. Its value is set as follows: car11, davidin use 0.9; car4, woman, shaking, animal, surfer use 0.95; trellis, singer1, bird2 use 0.99.

To reviewer 3:
Q: Justify: "robust features can be learned from auxiliary data and transferred to tracking target using deep learning", does the deep learning here help solve any intrinsic problems of visual tracking?
A: Since we just use a very simple classifier in the classification layer, the superiority of our method has to come from the feature extractor which corresponds to the encoder of the learned SDAE. Similar successes reported previously by others on many image and speech processing applications are also based on similar arguments. Our contribution could be seen as demonstrating the strengths of the deep learning approach in the context of an application to which deep learning has not been applied. Nevertheless, we agree that thoroughly addressing many application-specific issues related to visual tracking requires further research beyond our first attempt as reported here.

Q: "the features can be updated online to adapt to changing appearance while tracking". The authors just verbally argue its possibility, but do not implement it at all. Therefore, it becomes invalid.
A: Note that the feature extractor corresponds to the encoder of the SDAE (Fig 1(c)) which is updated by backpropagation during online tracking. So indeed we have implemented and demonstrated the efficacy of this idea. To make it more intuitive to understand by the reader, we plan to add a figure to show how the active features evolve during the tracking process.

Q: This paper is an incremental work.
A: As mentioned in #358-#363, though sharing a similar spirit, our model learns more generic features than [22], which only learns several predefined categories (Caltech 101). Moreover, their learned features are fixed during tracking while ours can further adapt to the objects which often experience appearance changes. While conceptually online adaptation sounds reasonable, making it to work efficiently and effectively does take quite a bit of effort. We have successfully demonstrated that this can be achieved by applying the deep learning approach appropriately.

Q: The efficiency of feature extraction should be considered as one of the strengths in this paper, but not compared to other methods under same settings.
A: As mentioned in #262, our method runs at about 2-20fps depending on the frequency of model update. We will include some timing statistics in the revised version of our paper.

Q: The weaknesses are not mentioned through the paper.
A: As mentioned in #364-#372, our model still tracks the target object using a holistic image. Patch-based methods are expected to produce more robust results against occlusion. We consider this extension as part of our future work.

Q: The reviewer suggests the authors present thorough comparisons with [22]
A: Thanks for the suggestion. Since the authors of [22] do not make their code available, direct comparison with their results is not possible.